# Spatio-temporal modelling for the evaluation of an altered Indian saline Ramsar site and its drivers for ecosystem management and restoration

**Rajashree Naik, Laxmikant Sharma** [ORCID]*

Department of Environmental Science, School of Earth Sciences, Central University of Rajasthan, Bandarsindri, Ajmer, Rajasthan, India

* laxmikant_evs@curaj.ac.in

**Data Availability Statement:** All relevant data are within the manuscript and its Supporting information files.

## Abstract

Saline lakes occupy 44% and 23% of the volume and area of all lakes that are tending to suffer from extended dryness, reduced hydro period, or complete desiccation by 2025. The current study is conducted on Sambhar Salt Lake, the largest inland saline Ramsar, site of India, contributing to 9.86% of total salt production. The lake is under threat due to illegal salt pan encroachment, losing brine worth 300 million USD. The objective was to identify the key drivers that affect the lake at a landscape level. Geospatial modelling was conducted for 96 years (1963–2059) at a decadal scale, integrating ground data (birds-soil-water). Land Use Land Cover (LULC) classification was conducted using CORONA aerial imagery of 1963, along with Landsat imageries, using supervised classification for 1972, 1981, 1992, 2009, and 2019, and future prediction for 2029, 2039, 2049, and 2059. Further, images were classified into 8 classes that include the Aravali hills, barren land, saline soil, salt crust, salt pans, wetland, settlement, and vegetation. Past trends show a reduction of wetland from 30.7 to 3.4% at a constant rate (4.23%) to saline soil, which subsequently seemed to increase by 9.3%, increasing thereby the barren land by 4.2%; salt pans by 6.6%, and settlement by 1.2% till 2019. Future predictions show loss of 40% wetland and 120% of saline soil and net increase in 30% vegetation, 40% settlement, 10% salt pan, 5% barren land, and a net loss of 20%, each by Aravali hills and salt crust. Additionally, the ground result shows its alteration and reduction of migratory birds from 3 million to 3000. In the light of UN Decade on Ecosystem Restoration (2021–2030), restoration strategies are suggested; if delayed, more restoration capital may be required than its revenue generation.

## Introduction

It is widely assumed that arid and semi-arid regions are devoid of water; however, they have numerous temporary and permanent water bodies. They also have high ecological, economic, cultural, recreational, and scientific values [1]. Surprisingly, these could be the alternatives for

**Funding:** The authors received no specific funding for this work.

**Competing interests:** The authors have declared that no competing interests exist.

freshwater sources, as globally, saline lakes make up 44% of the volume and 23% of the area of all lakes [2]. In contrast, these get little consideration due to their saline nature, and are thereby subject to water inflow diversions, construction of hydrological structures, pollution, mining, biological disruptions, and exotic species invasion [3]. Consequently, the hydro-patterns, water budget, hydrological communications, habitat alteration, loss of productivity, and connectivity among these lakes change [4]. These are also predicted to suffer from extended dryness, reduced hydro period, causing partial or complete desiccation by 2025 as already seen in Aral Sea, Lake Urmia, Owens Lake, Tarim Basin, and Salton Sea [2]. These cases have directly affected the billion-dollar global markets of shrimp, mineral industry, and caused ecological disruption [2]. Thus, it is critical to conduct systematic assessments of water resources in arid and semi-arid regions [5].

Earlier studies on saline lakes were challenging as they are mostly located in remote and inaccessible areas [6]. Eventually, the research on these lakes started with the physicochemical parameter assessments [6], phillipsite [7], chemical and biological properties [6], phytoplankton [8], primary productivity [9], stable isotopes [10], and geochemistry [11]. Until the last decades, most studies emphasized salinity as the factor responsible for biodiversity shifting in these lakes. However, the recent application of remote sensing (RS) and geospatial technology (GIS) have allowed landscape-level studies at a spatio-temporal scale [12]. These suggest lake size [13], habitat configuration [14], LULC [12] can also be the driving factor for the wetlandscape changes besides salinity. The next decade is likely to see the extensive use of habitat and niche modelling, climatic simulations, wetlandscape health assessment, past LULC trend analysis, and future predictions for these lakes using not only optical but also microwave, hyperspectral, LiDAR, Unmanned Aerial Vehicle (UAV) datasets integrated with machine learning, deep learning, Artificial Intelligence (AI) and Internet of Things (IoT). These are easy to standardize and competent to simulate even with small wetland complexes [15]. Therefore, this will rapidly enable the comprehension of sustainable management, restoration, combat desertification, biodiversity loss assessment and, water budgeting of these fast-degrading ecosystems [16].

Though the advancement in technology has enabled further research, studies have been mostly limited to large saline lakes (approximately 250 km$^2$ or more) like Great Salt Lake, Owens Lake, and Salton Sea of US; Aral Sea, Dead Sea, and Lake Urmia of Asia and Lake Chad of Africa. Additionally, there are also numerous unidentified small shallow saline lakes along with 390 permanent and temporary sites; out of 2400 Ramsar sites [17] which require urgent attention. India ranks third in the global salt market after China and the USA, contributing approximately 230 million tons, exporting to 198 countries. Some of the major importers include Bangladesh, Japan, Indonesia, South and North Korea, Qatar, Malaysia, U.A.E, and Vietnam. Moreover, in India, 96% of salt is produced from Gujarat, Tamil Nadu, and Rajasthan states with 76.7%, 11.16%, and 9.86% respectively from the sea, lake, sub-soil brine, and rock salt deposits. Specifically, the state of Rajasthan exports about 22.678 million tonnes of salt worth 340.17 billion dollars to the global market extracted from inland lake brine in Nawa, Kuchhaman, Rajas, Phalodi, Sujangarh, and Sambhar Salt Lake (SSL) [18]. The current study is conducted in Sambhar Salt Lake (SSL). It is the largest inland saline wetland ecosystem of India. It is also a gateway to the Thar Desert in India. It was designated as a Ramsar site on 23 March 1990, under criteria A with site No. 464 [19] and Important Bird Area (IBA) No. IN073 [20]. Once it was a haven for 279 migratory and resident birds [21], which currently serves as a refuge for 31 migratory only. Interestingly, despite many years of corruption, it is still not included under the country's protected network. The greatest threat to this lake is illegal encroachment in the core area [22]. Many illegal tube wells have been drilled, and long pumps have been used for groundwater over-extraction. Prior encroachment has turned it into a large

capital-intensive corporate business [23]. It is hard to ignore illegal consequences even after repeated National Green Tribunal (NGT) intervention [24].

Previous researches on SSL have only focused on birds [25], halo-tolerant species identification [26] and isolation [27], characterization [28, 29]; *Dunaliella sp.* [30] formation [31]; limnology [30]; paleoclimatology [29]; for sensor calibration and validation [32] and on extremophilic algal assessment [25]. However, the core challenge of the lake is its rapid shrinkage that cannot be addressed only by these time-consuming, tedious, and expensive methods. So, the objective of this article has been to identify the key drivers affecting SSL at a landscape level. We used integrated soil and water physico-chemical parameters, bird census data, with six decades of remote sensing data sets. The paper has been divided into five sections. Section one gives a brief overview of the saline lakes, the threats they are experiencing, and their future consequences. Section two describes the study area, data collection, and processing. Section three analyses the past fifty-six years of analysis (1963–2019), present (2019), and future forty years (up to 2059). Section four discusses the drivers of alteration, consequences, and restoration potentials and the last section provides the conclusion.

## Material and methods

### Study area

SSL is in the semi-arid climatic region of Rajasthan (Fig 1) with the geographical coordinates as 26° 52′ to 27° 02′ N; 74° 54′– 75° 14′ E running ENE–WSW direction in elliptical shape [20]. It is located 80.7 km away from Jaipur, the state capital via National Highway 48 and Rajasthan State Highway 57. In 1961, the Government of India (GoI) acquired the region on a 99-year lease under the Ministry of Commerce and Industry [19]. SSL is 230 km$^2$ (22.5 km in length and 3–1 km in width) [26]. One of the world's oldest mountain ranges, Aravali hills surrounds it in the north, west, and south-east directions, extending up to 700 m [20]. Its maximum altitude is 360 m above mean sea level, with 0.1 m per 1000 m slope. Ephemeral streams (Mendha, Rupnagar, Khandel, Kharian) form the catchment of 5,520 km$^2$. Mendha in fact is the largest feeder river that originates in the north from Sikar district and drains out in 3600 km$^2$. Notably, it experiences a tropical climate, and its soil consists of silt and clay. Some part of the basin is calcareous, while most of it is argillaceous; it is rich in salts of sodium, potassium, calcium, and magnesium cations and carbonate, bicarbonate, chloride, and sulphate anions. It appears white especially in areas with rich salt content; appears grey in areas with less salt content, and brown with no salt content [33]. Importantly, SSL at large, experiences distinct summer (March-June), rainy seasons (July-September), and winter (October-February). Overall, it receives around 500 mm of rainfall every year, while it enjoys 250–300 sunny days [20]. Additionally, the average temperature is about 24.4˚C, going up to 40.7˚C in summer, and below 11˚C in winter [33]. Further, during rainy seasons, it looks like a muddy blackish wetland [28]. It has almost 3 m depth during monsoons but shallows down to 0.6 m during dry periods. Except for reservoir and salt pans, the whole lake dries up exposing salt flakes during summer [34]. A 5.16 km long dam divides into two unequal parts (77 km$^2$ towards the east as a reservoir and the rest 113 km$^2$ is wetland) [29]. It receives migratory birds of Central Asian, East Asian, and East African flyways. Invertebrates, amphibians, crustaceans come through rivers during the monsoon when salinity is low [26]. Moreover, it provides shelter to 37 herbs (*Portulaca oleracea, Salsola foetida, Suaeda fruticose*) 14 shrubs (*Salvadora oleoides, Salvadora persica, Sericotoma pauciflorum*) 14 trees (*Acacia nilotical, Acacia Senegal, Anogeissus pendula*) 15 grasses (*Apluda mutica, Aristida adscensionis, Cenchrus ciliaris*) 6 chlorophycea (*Chamydomonas sp., Dunelialla salina, Oedogonium sp.*) 25 Cyanophyceae (*Lyngbya sp.,*

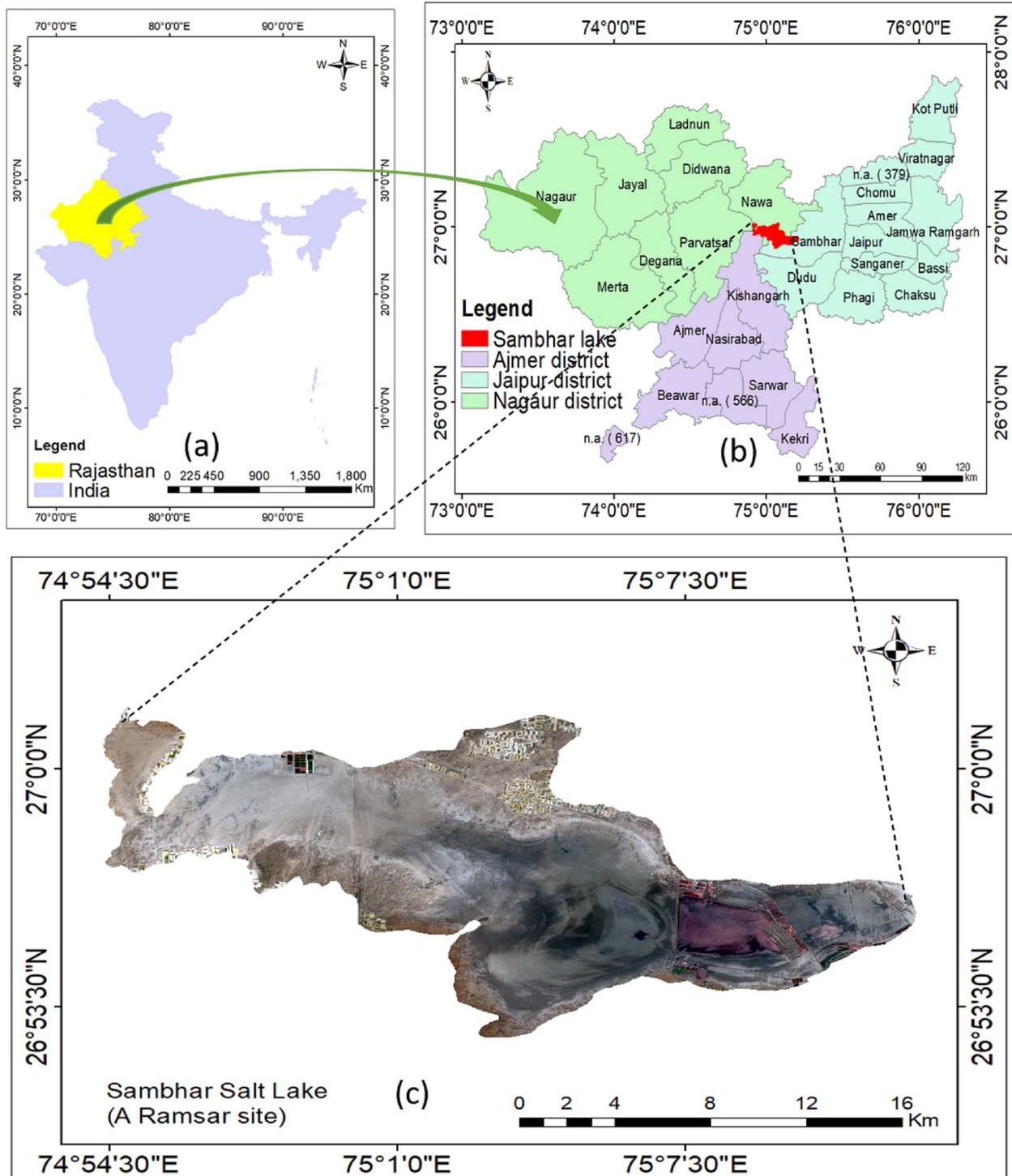

**Fig 1. Study area.** (a) India with Rajasthan state highlighted in yellow colour (b) SSL amid Nagaur, Jaipur and Ajmer districts (c) True Colour Composite of SSL prepared using Sentinel data set of 7 May 2021.

*Merismopedia sp*.,*Microcole,us sp.)* and 7 Bacilariophyceae *(Cymbella sp*., *Melosira sp*., *Navicula sp)* species [35].

## Data processing

Four types of data have been used in this study. These are soil, water, birds, and remote sensing data. The study also has used an integrated research design combining all these datasets. Before the experiment started, a field visit was conducted for prior knowledge and protocol [36] was developed. Then, the experiment was conducted in two phases. In the first phase, satellite data were collected, processed, and analysed. In the second phase, a sampling design was prepared, subsequently, soil-water samples were collected and analysed in the laboratory, and bird census was conducted. All the experiments were done in winter. As birds visit during the winter season, unlike summer, when the lake has consistent water level, feeding and breeding facilities for them. So, identification of the drivers of alteration would be more accurate during this season.

LULC classification was carried out for past, current, and future changes on a decadal scale. Notably, only one aerial image of CORONA was obtained, which is well before the start of any satellite programmes. This photograph has only been digitized for visual interpretation and area calculated for LULC classes; area calculated, however, as this is a high-resolution image, the calculated area could not be comparable with other satellite datasets. Satellite data including Landsat- Multispectral Scanner System (MSS) of 16 November 1972 and 18 October 1981, Landsat-5 Thematic Mapper (TM) of 25 November 1992 and 8 November 2009, and Landsat-8 Operational Land Imager (OLI) of 20 January 2019 were also collected. Images of 1972 and 1981 have 60 m spatial resolution, while the rest of the images have 3 M resolution. The best available cloud-free images of the Landsat satellite were downloaded within this season. Additionally, the study was not impacted by droughts or floods. The study years, 1972, 1981, 1992 received rainfall above 500 mm, while 2009 and 2019 received just about average rainfall [36]. The methodology followed has been shown (Fig 2).

All of the downloaded images were geo-referenced and pre-processed, including atmospheric and geometric correction. Pre-processing is a necessary step for Landsat data due to existing instrumental errors, geometric and scale uncertainty, and different noise of the sensors MSS, TM, ETM, ETM+, and OLI. To maintain uniformity, the pan-sharpening of 1972 and 1981 images was done to 30 m spatial resolution. Toposheet from Survey of India (1954) at 1:26,000 scale was used for boundary delineation. SSL was digitized and a 3 km buffer was selected, as it is declared as an eco-sensitive zone, according to Rajasthan State Forest Department. For classification, the pixel-based method was used using ERDAS Imagine, 2014, while the final maps were composed using Arc GIS 10.5. Further, SSL was divided into eight classes using the supervised classification method; they include wetland, salt pan, salt crust, vegetation, the Aravali hills Mountain range, saline soil, barren land, and settlement. The water bodies represent the wetland areas, which do not come under the reservoir. It appears dark-light blue in True Colour Composite (TCC). Salt pans, on the other hand, are the salt-producing units; salt crust represents high salt deposition area, appearing white, while vegetation appears green, and are occupied by both xerophytes and halophytes, Aravali hills represent the hill ranges, saline soil represents the terrestrial part of the lake with both soil and salt content appearing grey, barren land represents area without salt content appearing brown, while settlement represents the built-up area surrounding SSL. Moreover, past change detection was conducted for 47 years (1972–2019) on a decadal scale. Notably, the LULC of each image was estimated using pixel-based classification. Supervised Maximum Likelihood classification method (MLC) was applied. 69 GPS locations were obtained from in and around the lake

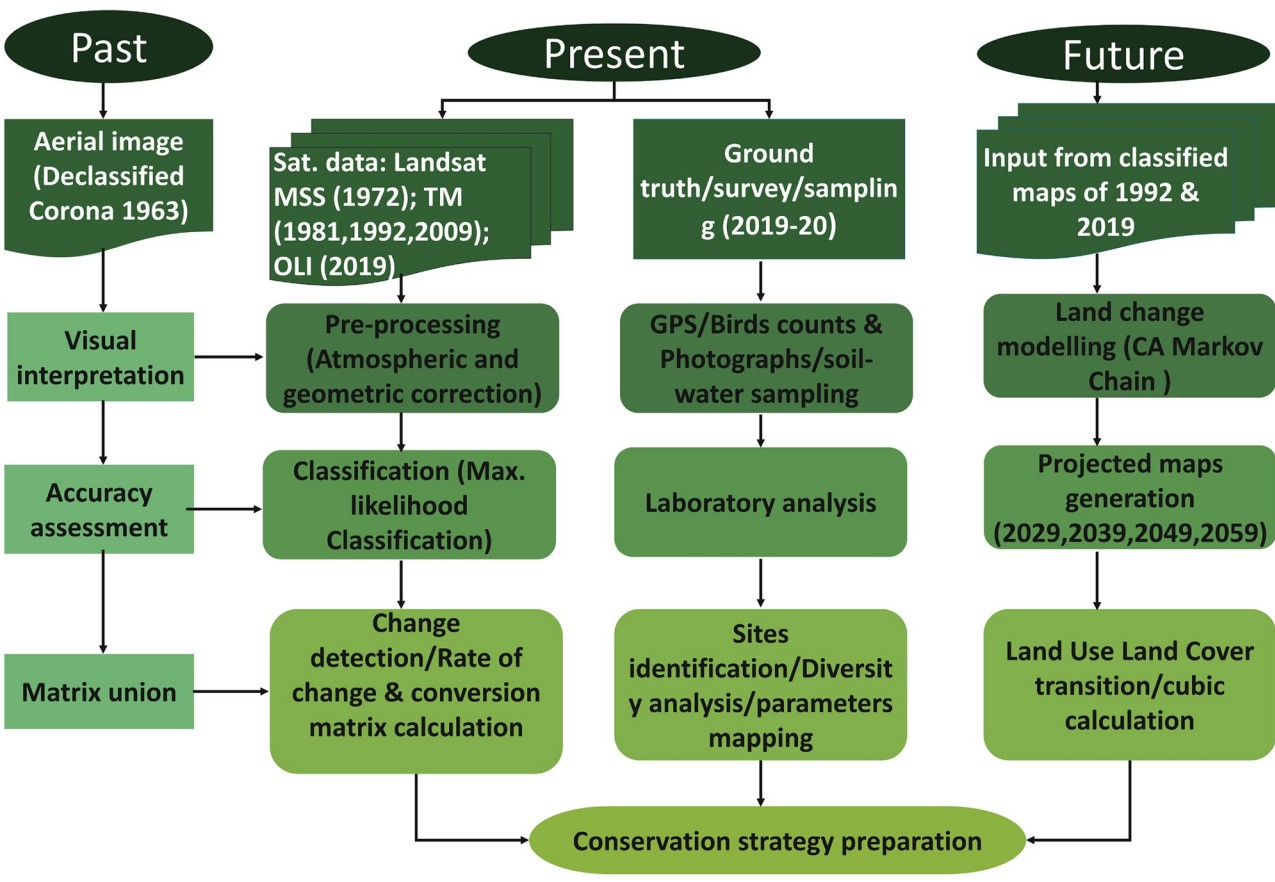

**Fig 2. Methodology flowchart.**

during soil and water sample collections, bird census, and validation of classification shown (Fig 3).

Target locations were pre-defined for each class and sampling. Detailed field research was carried out on 13 February 2019 (winter); 10 April 2019 (summer); 30 June 2019 (monsoon), and 6 January 2020. Additionally, out of 69 GPS locations, 48 points were used for classifications and other points for accuracy assessment. Primary landmarks like historical sites (Shakambari temple, Devyani Sarovar, Dadu Dayal point, Sambhar City, Railway Station), along with birding sites, dumping sites, illegal pans, sewage points, tourist construction sites, salt processing, and packaging sites were identified (Fig 4a–4f) and historical photographs were collected from [37] as given (Fig 4g and 4h). Google Earth was used for accuracy assessment of the past images.

The dynamic degree was calculated using [38]

$$K = \frac{Ub - Ua}{Ua} * \frac{1}{T} * 100\% \tag{1}$$

K is the land use dynamic degree, calculated as percent LULC change per year, both Ua and Ub represent areas under a specific annual LULC, while T represents the time in years. Dynamic matrix was generated between the years 1981–2019 to estimate LULC transfer. Notably, to quantify the transition matrix thematically, an equal number of classes are required.

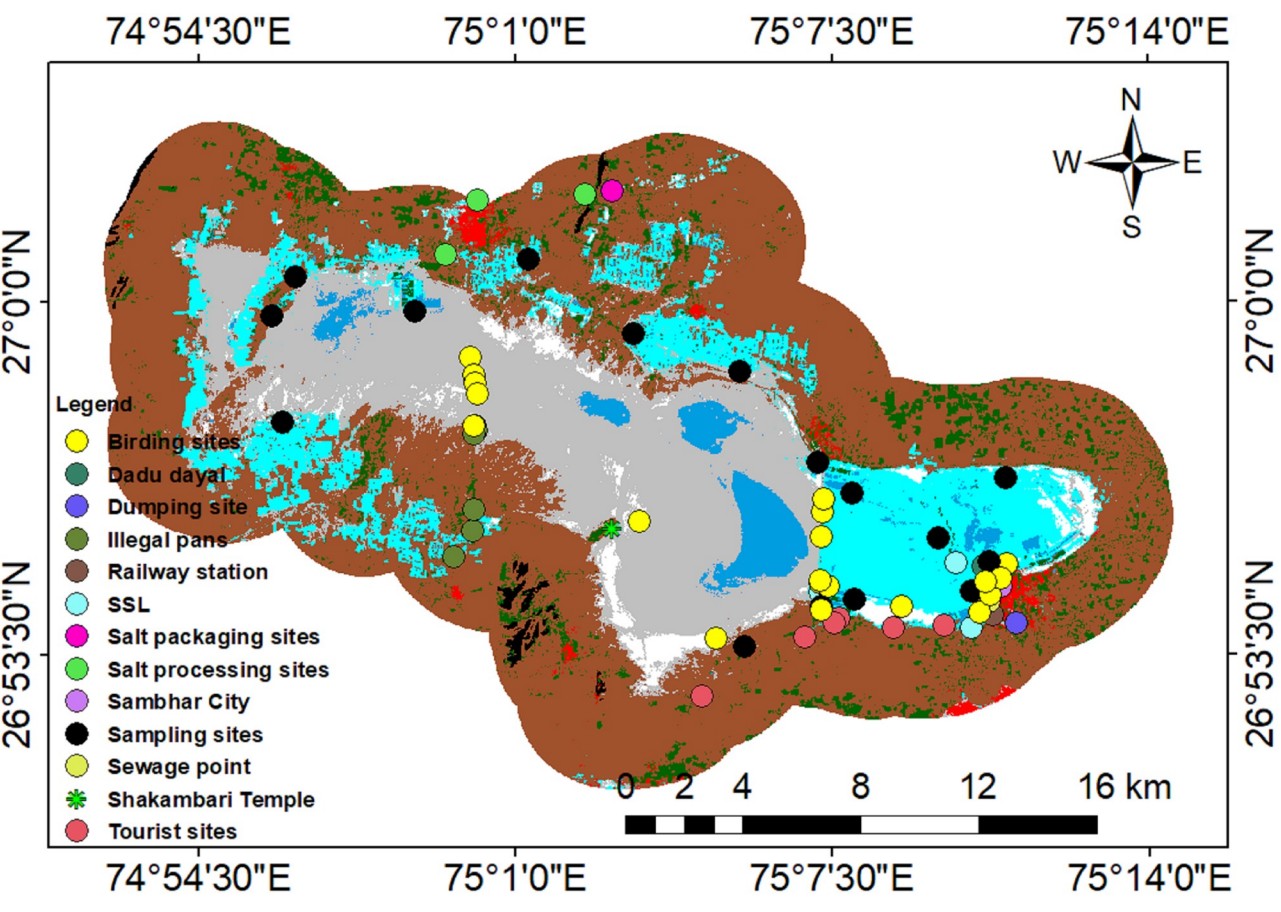

**Fig 3. Location of field points.**

Classified image of 1981 was taken instead of 1972 for matrix analysis, as SSL was subject to flash floods caused by 1000 mm of rainfall in 1971. So, due to rising water levels, no salt crust was observed in the raw image. For future projection of four decades (from 2029–2059), the Cellular Automata (CA) Markov Chain Model of Land Change Modelling (LCM) was conducted using Terra Set software. Classified images of 1992, 2009, and 2019 (two decades representatives) were used for forecasting. The quantitative results were achieved using LCM, the net change of each class was calculated and then the factors of change were calculated.

Moreover, to assess the current situation, three parameters (i.e., soil, water, and bird count) were chosen. Sixteen samples, each for soil and water were collected by stratified random sampling method. These collected samples were further analysed in the laboratory of the Department of Environmental Science as per American Public Health Associations (APHA) guidelines. pH and electrical conductivity were examined using the respective electrode. Other parameters like salinity, chloride, carbonate, total organic carbon of soil and total dissolved solids, salinity, hardness, carbonate of samples were analysed using the titration method.

Bird censuses were conducted on 11 January, 2019, and on 6th and 7th January 2020. From the survey, 29 and 32 bird species with a total of 1124 and 43,445 bird count were recorded in the respective years (Table 1). Good rainfall increased bird counts in 2019. 10 bird species like Black Crowned Night Heron, Greater Flamingo, Lesser Flamingo, Gadwall, Little Ringed Plover, Kentish Plover, Red Wattled Lapwing, Black Winged Stilt, Pier Avocet, and Common

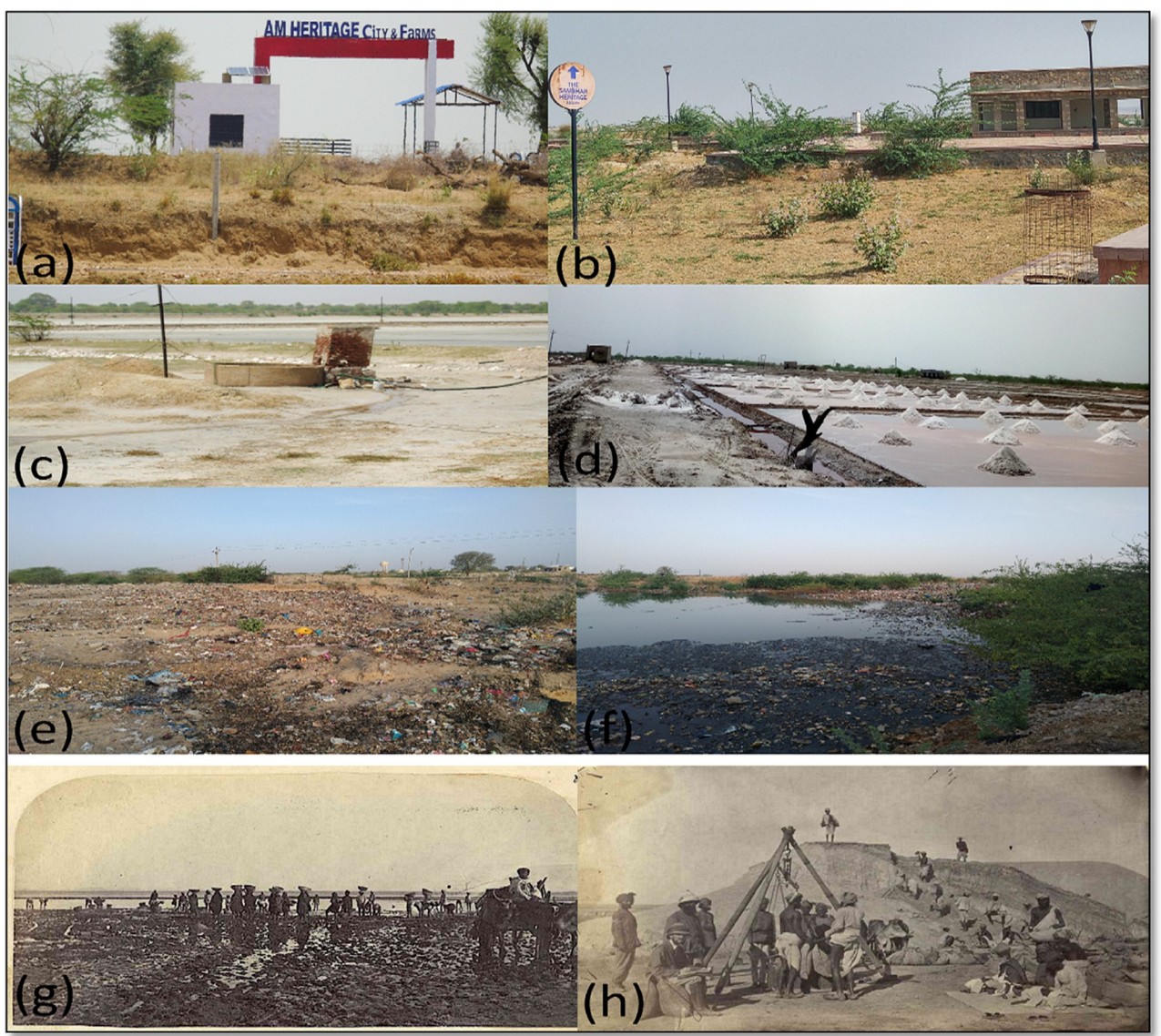

**Fig 4. Field photographs.** (a) & (b) construction sites, (c) & (d) surface wells, salt pans with illegal electrical cables (e) & (f) domestic pollution sources, and (g) & (h) historical photographs of lake and salt production.

Sandpiper have a strong preference for saline and alkaline lakes that attracts them to SSL. Importantly, some species feeding upon invertebrates little Grebe, Graylag Goose, Bar-Headed Goose, Common Teal, Northern Shoveler, Great Stone Plover, White-Tailed Lapwing, Black-tailed Godwit, Common Redshank, Curlew Sandpiper, Marsh Sandpiper, Wood Sandpiper, Little Stint, Temmick's Stint, Ruff, White Wagtail, Grey Wagtail, Pin-tailed Snipe, and Yellow wattled Lapwing found in and around SSL. A species-wise detailed bird census conducted by [21] stated that a total of 83 waterfowls was recorded. In 1994, 8,500 lesser flamingos were seen on the lake, but no greater flamingos were found; in 1995, 5,000 lesser flamingos were recorded but no greater flamingos were observed, in 2001, 20,000 birds were observed out of which 10,000 were lesser and 5,000 were greater flamingos. The absence of birds is indicated as Not Found (NF) in the table.

**Table 1. Waterbirds counts and comparison.**

| S No. | Common and scientific name | 2019 | 2020 |
|---|---|---|---|
| | Grebes | | |
| 1 | Little Grebe *Tachybaptus ruficollis* | 9 | NF |
| | HERONS, EGRETS and BITTERNS: | | |
| 2 | Black-crowned Night Heron *Nycticorax nycticorax* | 3 | NF |
| 3 | Indian Pond Heron *Ardeola grayii* | NF | 1 |
| 4 | Cattle Egret *Bubulcus ibis* | NF | 5 |
| | Flamingos: | | |
| 5 | Greater Flamingo *Phoeniconaias* | 331 | 12,046 |
| 6 | Lesser Flamingo | 128 | 24,413 |
| | GEESE and DUCKS: | | |
| 7 | Greylag Goose *Anser anser* | 6 | NF |
| 8 | Barheaded Goose *A. indicus* | 18 | NF |
| 9 | Common Pochard *Aythya ferina* | NF | 3 |
| 10 | Gadwall *A. strepera* | 10 | NF |
| 11 | Common Teal *A. crecca* | 9 | NF |
| 12 | Northern Shoveler *A. clypeata* | 359 | 5,293 |
| | GULLS, TERNS and SKIMMERS: | | |
| 13 | Brown-headed Gull *L. brunnicephalus* | NF | 1 |
| | Plovers: | | |
| 14 | Great Stone Plover *Esacus recurvirostris* | 1 | NF |
| 15 | Little-ringed plover *C. dubius* | 4 | 25 |
| 16 | Pacific Golden Plover | NF | 1 |
| 17 | Kentish Plover *C. alexandrinus* | 4 | 47 |
| 18 | Red-wattled Lapwing *V. indicus* | 12 | 16 |
| 19 | White-tailed Lapwing *V. leucurus* | 1 | 2 |
| | Stilts, avocets: | | |
| 20 | Black-winged Stilt *Himantopus himantopus* | 16 | 112 |
| 21 | Pied Avocet *Recurvirostra avosetta* | 34 | 422 |
| | Snipes, curlews, sandpipers, shanks, godwits, stints: | | |
| 22 | Black-tailed Godwit *Limosa limosa* | 2 | 2 |
| 23 | Eurasian Curlew *N. arquata* | NF | 1 |
| 24 | Common Redshank *T. totanus* | 3 | 25 |
| 25 | Common Greenshank *T. nebularia* | NF | 5 |
| 26 | Curlew Sandpiper *C. ferruginea* | 26 | NF |
| 27 | Marsh Sandpiper *T. stagnatilis* | 1 | 2 |
| 28 | Green Sandpiper *T. ochropus* | NF | 2 |
| 29 | Wood Sandpiper *T. glareola* | 5 | 7 |
| 30 | Common Sandpiper *Actitis hypoleucos* | 2 | 15 |
| 31 | Little Stint *C. minuta* | 8 | 110 |
| 32 | Temminck's Stint *C. Temminckii* | 17 | 9 |
| 33 | Ruff *Philomachus pugnax* | 140 | 441 |
| | Kingfishers: | | |
| 34 | White-breasted Kingfisher *H. smyrnens* | NF | 3 |
| | EAGLES, OSPREY, HARRIERS, FALCONS, KITES: | | |
| 35 | Western Marsh-Harrier *Circus aeruginosus* | NF | 1 |
| | WAGTAILS, PIPIT: | | |
| 36 | White Wagtail *Motacilla alba* | 2 | 3 |

(*Continued*)

**Table 1.** (Continued)

| S No. | Common and scientific name | 2019 | 2020 |
|---|---|---|---|
| 37 | White-browed Wagtail *M. maderaspatensis* | NF | 1 |
| 38 | Grey Wagtail *M. cinerea* | 1 | |
| | Additional species | | |
| 39 | Pacific Golden Plover | NF | 1 |
| 40 | Raptor | NF | 3 |
| 41 | Crested Lark | NF | 5 |
| 42 | Greater Coual | NF | 1 |
| 43 | Pintailed Snipe | 5 | NF |
| 44 | Yellow lapwing | 1 | NF |
| 45 | Undefined | NF | 422 |
| | Total count | 1124 | 43,445 |
| | Total species no. | 29 | 32 |

## Results

### Past 56 years

Visual interpretation of CORONA revealed four geomorphic units (Aravali hills, rivers, saline soil, and lake). Two major rivers were identified due to their shape, Mendha in the north and Rupnagar in the south with their rivulets. Bright tone and smooth textured landform are saline soil, which has further reduced in subsequent years (Fig 5). In 1963, the area of SSL was 228.80 km$^2$, Aravali hills were 11.11 km$^2$, the settlement was 3.40 km$^2$, saline soil was 0.15 km$^2$, the salt pan was 39.54 km$^2$, vegetation was 5.80 km$^2$, and barren land was 214.66 km$^2$. Importantly, the image of 1972 was classified at 80.95% accuracy with 0.73 Kappa coefficients, 1981 at 82.50% with 0.76, 1992 at 87.50% with 0.82, 2009 at 85.71% with 0.80 and 2019 at 87.50% at 0.82. This accuracy is reflected in the analysis of the dynamic matrix.

The study shows the degraded trend of the area of LULC for 47 years between 1972–2019 (Table 2, Fig 6). In 1972, the wetland was 159.6 km$^2$ (30%), salt pan was 38.3 km$^2$ (7.4%), salt crust was 0 km$^2$ (0%), vegetation was 17.9 km$^2$ (3.4%), Aravali hills was 3.5 km$^2$ (0.7%), saline soil was 64.3 km$^2$ (12.4%) and barren land was 236.0 km$^2$ (45.4%). In 1981, wetland was 98.7 km$^2$ (19%), salt pan was 36.1 km$^2$ (6.9 km2), salt crust was 34.4 km$^2$ (6.6%), vegetation as 87.6 km$^2$ (16.9%), Aravali hills was 3.3 km$^2$ (0.6%), saline soil was 49.1 km$^2$ (9.4%), barren land was 209.6 km$^2$ (40.3%) and settlement was 1.1 km$^2$ (0.2%). In 1992, wetland was 106.7 km$^2$ (20.5%), salt pan was 42.8 km$^2$ (8.2%), salt crust was 34.7 km$^2$ (6.7%), vegetation was 5.3 km$^2$ (1.0%), Aravali hills was 3.3 km$^2$ (0.6%), saline soil was 90.7 km$^2$ (17.5%), barren land was 235.3 km$^2$ (45.2%) and settlement was 1.1 km$^2$ (0.2%). In 2009, wetland was 31.5 km$^2$ (6.1%), salt pan was 64.1 km$^2$ (12.3%), salt crust was 0.0 km$^2$ (0%), vegetation was 84.1 km$^2$ (16.2%), Aravali hills was 3.2 km$^2$ (0.6%), saline soil was 118.3 km$^2$ (27.7%), barren land was 217.3 km$^2$ (41.8%) and settlement was 1.4 km$^2$ (0.3%). In 2019, wetland was 17.4 km$^2$ (3.4%), salt pan was 72.9 km$^2$ (14.0%), salt crust was 15.4 km$^2$ (3.0), vegetation was 34.1 km$^2$ (6.6%), Aravali hills was 3.2 km$^2$ (0.6%), saline soil was 112.6 km$^2$ (21.7%), barren land was 257.8 km$^2$ (49.6%) and settlement was 6.5 km$^2$ (1.3%). Overall, the change from 1972 to 2019 has been summarized, as wetland decreased from 30.7 to 3.4%. Salt crust increased from 0 to 3%. Vegetation increased from 3.4 to 6.6%. Aravali hills decreased from 0.7 to 0.6%. Saline soil increased from 12.4 to 21.7%. Barren land increased from 45.4 to 49.6%. Salt pan increased from 7.4 to 14%. Settlement increased from 0.1 to 1.3%.

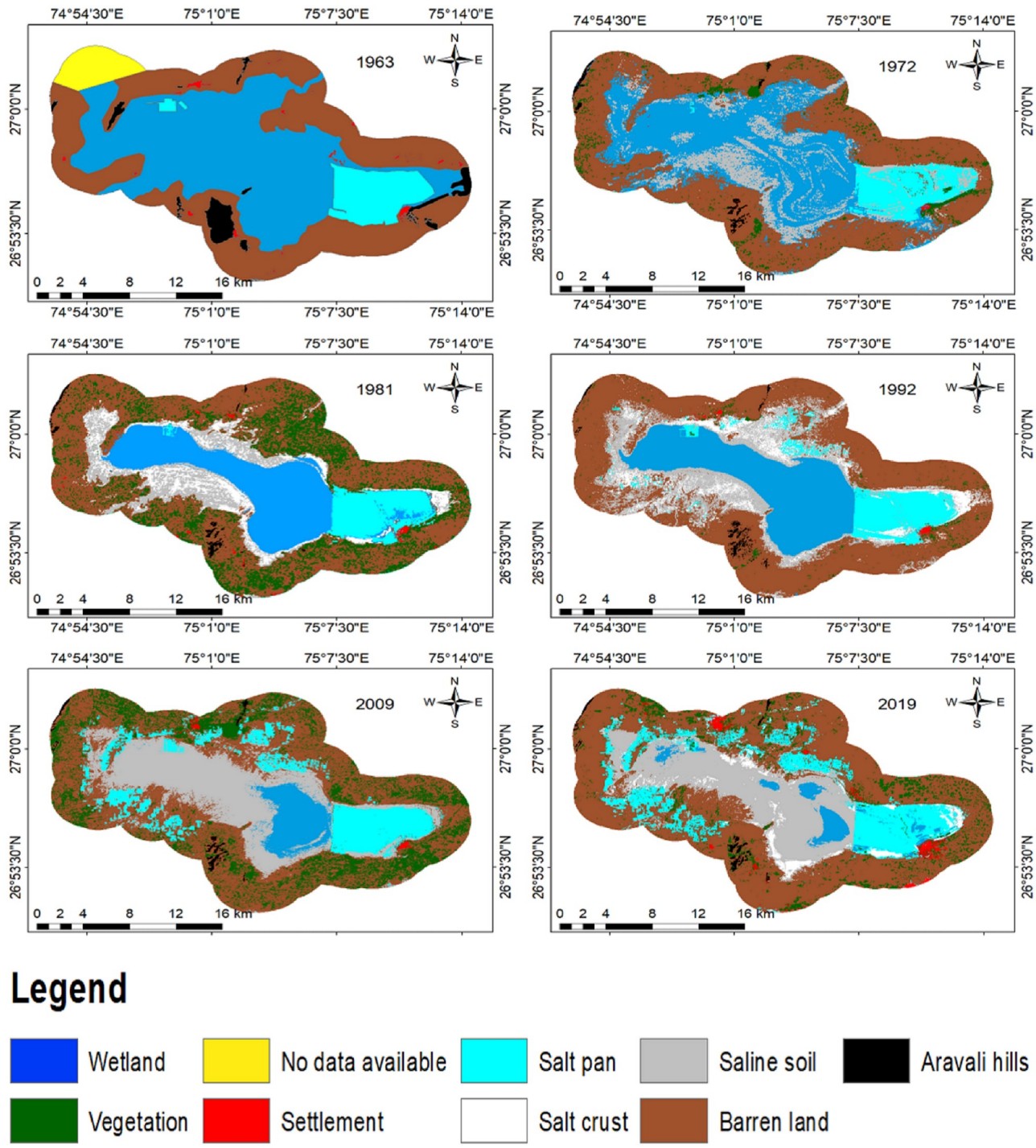

**Fig 5. Past LULC maps.** 1963, 1972, 1981, 1992, 2009 and 2019 of SSL.

LULC change rate (Table 3) is represented as K (%). It shows the wetland degrading at the rate of -4.23%, 0.73%, -4.14%, and -4.47% since 1972–2019. In fact, in the first decade, K of vegetation was 43.38% and settlement was 12.66%. Salt pan decreased by 0.63%, Aravali hills by 0.56%, saline soil by 2.62%, and barren land by 1.24%. Furthermore, from 1981 to 1992,

**Table 2. LULC change area from 1972–2019 (area in km$^2$).**

| LULC | 1972 | | 1981 | | 1992 | | 2009 | | 2019 | |
|---|---|---|---|---|---|---|---|---|---|---|
| | Area | % | Area | % | Area | % | Area | % | Area | % |
| Wetland | 159.6 | 30.7 | 98.7 | 19.0 | 106.7 | 20.5 | 31.5 | 6.1 | 17.4 | 3.4 |
| Salt pan | 38.3 | 7.4 | 36.1 | 6.9 | 42.8 | 8.2 | 64.1 | 12.3 | 72.9 | 14.0 |
| Salt crust | 0 | 0.0 | 34.4 | 6.6 | 34.7 | 6.7 | 0.0 | 0.0 | 15.4 | 3.0 |
| Vegetation | 17.9 | 3.4 | 87.6 | 16.9 | 5.3 | 1.0 | 84.1 | 16.2 | 34.1 | 6.6 |
| Aravali hills | 3.5 | 0.7 | 3.3 | 0.6 | 3.3 | 0.6 | 3.2 | 0.6 | 3.2 | 0.6 |
| Saline soil | 64.3 | 12.4 | 49.1 | 9.4 | 90.7 | 17.5 | 118.3 | 22.7 | 112.6 | 21.7 |
| Barren land | 236.0 | 45.4 | 209.6 | 40.3 | 235.3 | 45.2 | 217.3 | 41.8 | 257.8 | 49.6 |
| Settlement | 0.5 | 0.1 | 1.1 | 0.2 | 1.1 | 0.2 | 1.4 | 0.3 | 6.5 | 1.3 |

only vegetation changed negatively; the rest of the classes like increased wetland by 0.73%, salt pan by 1.68%, salt crust by 0.08%, Aravali hills by 0.06%, saline soil by 7.71%, and barren land by 1.11% and settlement by 0.43%. From 1992 to 2009, wetland decreased by -4.14% followed by salt crust by 5.88%, Aravali hills by 0.11%, and barren land by 0.45% whereas vegetation by 0.20%, and saline soil by 1.78% positive K. From 2009 to 2019, wetland, vegetation, Aravali hills, salt crust, and saline soil showed negative K by 4.47%, 5.95%, 0.11%, 0.00%, and 0.48% respectively and salt pan, barren land, and settlement showed positive K of 1.36%, 1.86%, and 37.98% respectively. The settlement has high K value in this decade.

LULC transition matrix (Table 4) states conversion from wetland (75 km$^2$) to saline soil, second largest from barren land to (22.5 km$^2$) to vegetation. 0.96 km$^2$ of Aravali hills to barren land, saline soil (21.67 km$^2$) to barren land. 13.87 km$^2$ and 12.11 km$^2$ of salt crust are to saline soil and barren land respectively.

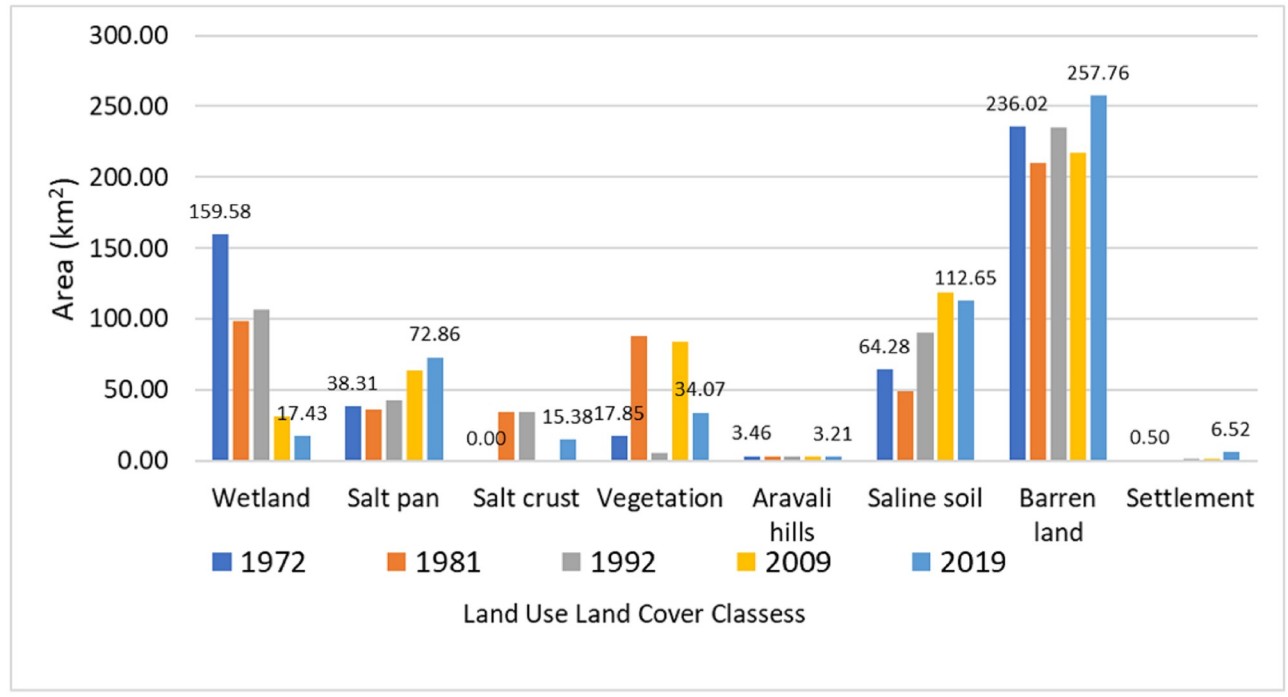

**Fig 6. Graph of LULC change.**

**Table 3. LULC dynamic degree K (percentage).**

| LULC classes | 1972–81 | 1981–92 | 1992–09 | 2009–19 |
|---|---|---|---|---|
| Wetland | -4.23% | 0.73% | -4.14% | -4.47% |
| Salt pan | -0.63% | 1.68% | 2.93% | 1.36% |
| Salt crust | 0.00% | 0.08% | -5.88% | 0.00% |
| Vegetation | 43.38% | -8.54% | 88.20% | -5.95% |
| Aravali hills | -0.56% | 0.06% | -0.11% | -0.11% |
| Saline soil | -2.62% | 7.71% | 1.78% | -0.48% |
| Barren land | -1.24% | 1.11% | -0.45% | 1.86% |
| Settlement | 12.66% | 0.43% | 1.25% | 37.98% |

## Current status of SSL

**Soil and water quality parameters.** Soil parameters like pH, Electrical Conductivity (EC), salinity, chloride, carbonate, and Total Organic Carbon (TOC) were analysed and mapped (Fig 7). Linear correlation was calculated. The highest positive correlation was observed between salinity and EC (r = 0.99), indicating salts as the major factor for conductivity. Anions like chloride, carbonates, and bicarbonates, chloride have very high EC (r = 0.99), which infers those chlorides are major anions responsible for salinity. TOC is slightly positively related (r = 0.5) to other parameters. Water parameters like pH, EC, Total Dissolved Solids (TDS), salinity, chloride, carbonate, and hardness were calculated and mapped (Fig 8). Linear correlation was calculated. The highest positive correlation was observed between salinity and EC (r = 0.93) and the same positive relationship between salinity and TDS. As per the analysis, major salinity has been contributed by chloride ions; it also shows a positive correlation between EC and TDS. Other than the chloride, carbonate has also a correlation with TDS (r = 0.5).

## Future LULC for next 40 years

The future prediction was conducted. Predicted maps of 2029, 2039, 2049, and 2059 were obtained (Fig 9). There will be a decrease in wetland area and increases in salt pans towards north. Conversion of saline soil into barren land in the central part is observed. Towards the south, salt pans will have no noticeable change until 2039; however, will increase in 2049 and 2059 maps. Area statistics graphs for predicted maps were derived through modelling (Fig 10a–10j). (a) Shows percentage-wise gain and loss; (b) shows the net change in percentage and (c to j) shows class-wise contributions to net changes.

**Table 4. LULC transition matrix from 1981–2019.**

| 1981 | 2019 | | | | | | | | |
|---|---|---|---|---|---|---|---|---|---|
| | Aravali hills | Barren land | Saline soil | Salt crust | Salt pan | Settlement | Vegetation | Water body | Grand Total |
| Aravali hills | 2.12 | 0.96 | 0.00 | 0.00 | 0.01 | 0.01 | 0.11 | 0.01 | 3.23 |
| Barren land | 1.02 | 155.46 | 3.54 | 1.04 | 21.26 | 3.25 | 22.57 | 0.55 | 208.69 |
| Saline soil | 0.00 | 21.67 | 17.86 | 2.96 | 4.47 | 0.04 | 1.97 | 0.10 | 49.08 |
| Salt crust | 0.00 | 12.11 | 13.87 | 4.33 | 2.99 | 0.10 | 0.95 | 0.08 | 34.43 |
| Salt pan | 0.00 | 0.89 | 0.32 | 1.92 | 29.21 | 0.33 | 0.96 | 2.41 | 36.04 |
| Settlement | 0.08 | 66.07 | 1.22 | 0.26 | 11.63 | 2.78 | 7.35 | 0.40 | 89.79 |
| Wetland | 0.00 | 0.70 | 75.84 | 4.86 | 3.30 | 0.02 | 0.16 | 13.87 | 98.74 |
| Grand Total | 3.22 | 257.85 | 112.65 | 15.38 | 72.86 | 6.52 | 34.09 | 17.43 | 520.01 |

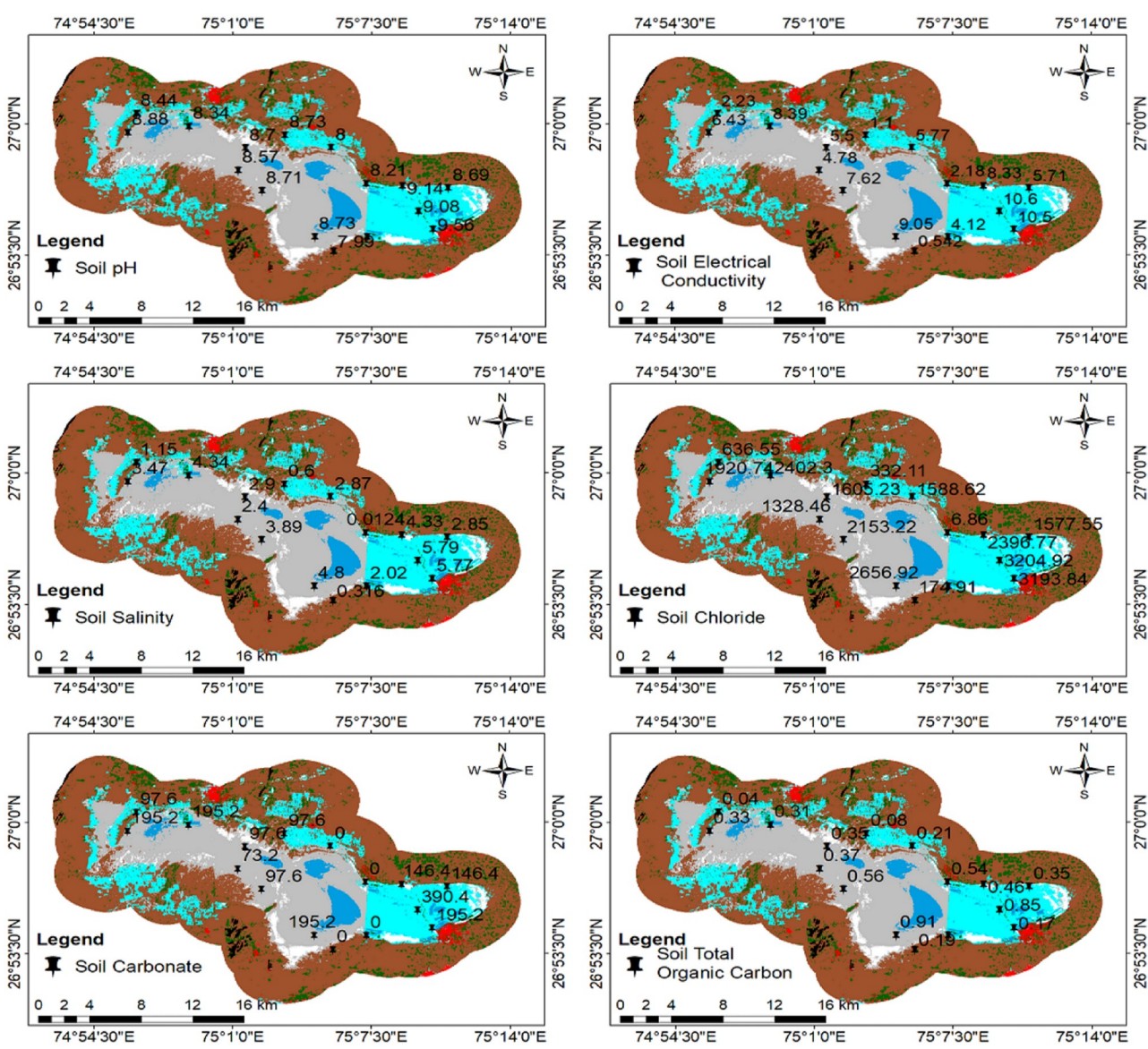

**Fig 7. Mapping soil quality parameters.**

Fig 10(a) shows the highest gain in the wetland by 60% and the highest loss of saline soil by 70%. It shows wetland loss by 40%, vegetation gained 40% and loss 30%, the settlement gained by 50% and loss by 20%, salt pan gain by 30% and loss by 20%, salt crust gain by 40%, and loss by 50%, saline soil gain by 40% and loss by 70%, barren land gain by 20% and loss by 10% and Aravali hills gain by 40% and loss by 20%. Fig 10(b) shows net increase by 20% in wetland, 30% in vegetation, 40% in the settlement, 10% in the salt pan, 5% in a barren land, and a decrease by 20% in a salt crust, saline soil by 120%, and Aravali hills by 20%. Fig 10(c) to 10(j) shows the net change in each LULC. Aravali hills will positively impact by 0.01% and negatively by 0.02% and 0.04% by settlement and barren land. Positive contributions to a net change of barren land by salt crust and saline soil are by 0.01% and 5% respectively, negatively in the wetland by 0.01%, vegetation by 0.5%, settlement by 0.05%, and salt pan by 3%. Positive

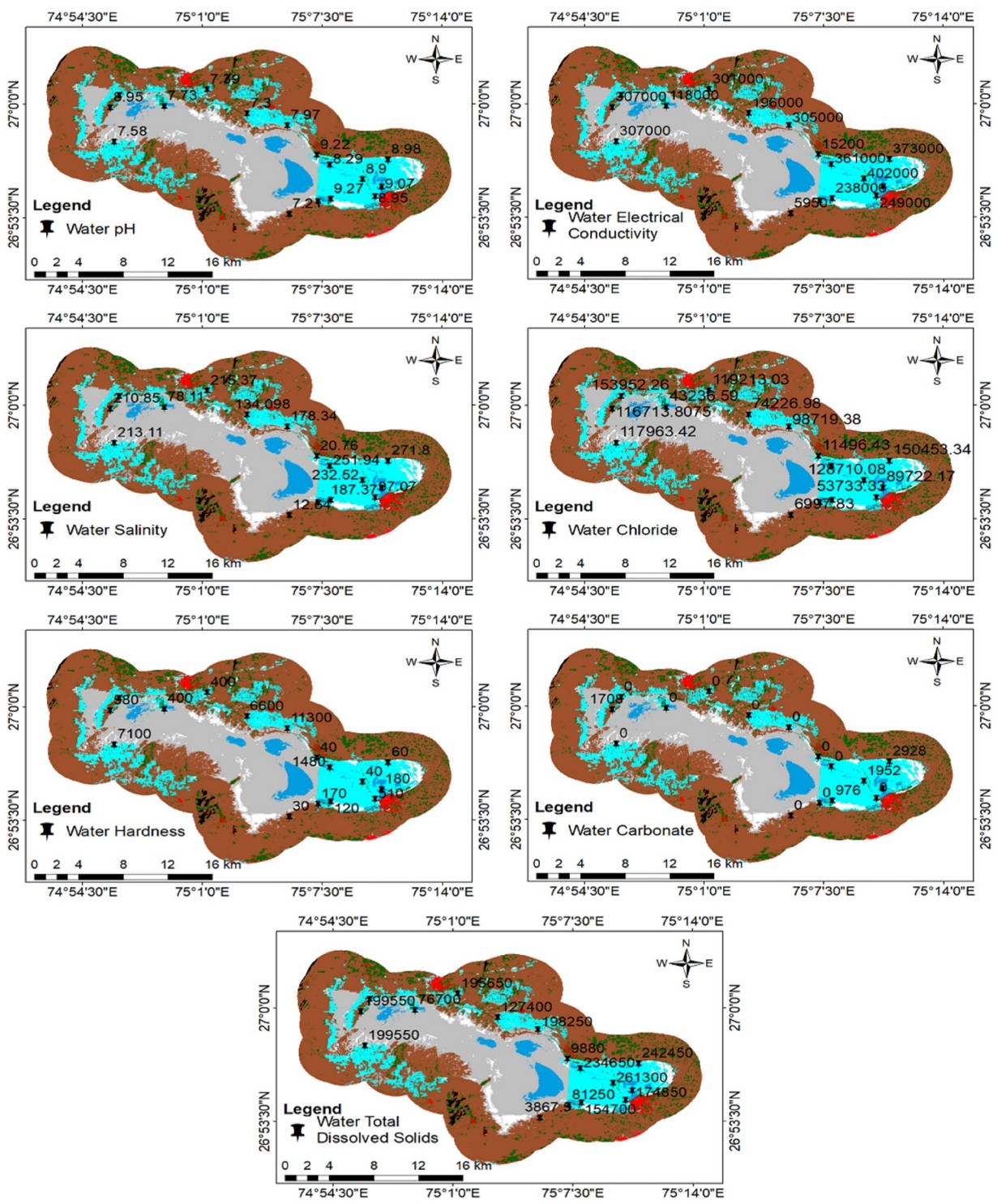

**Fig 8. Mapping water quality parameters.**

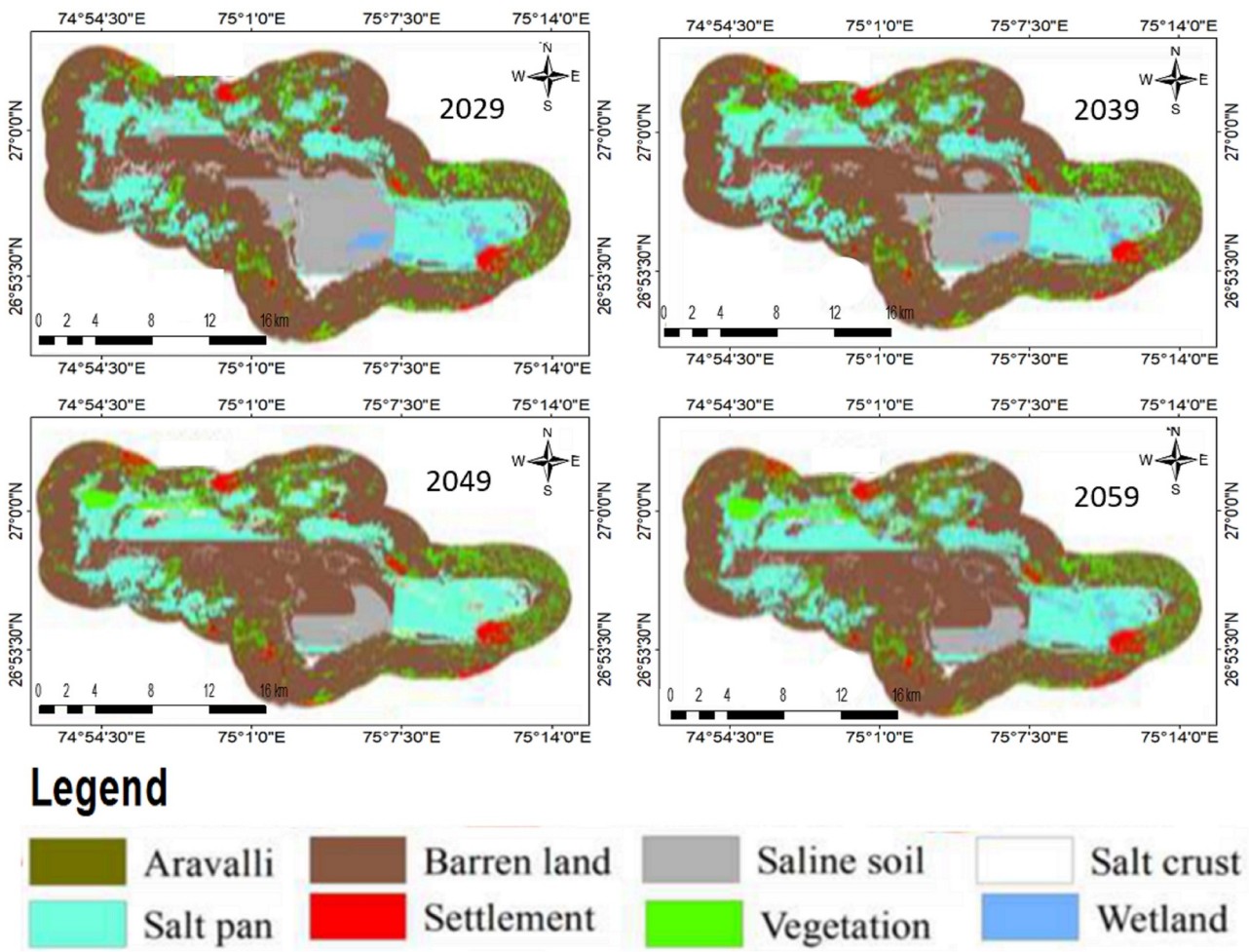

**Fig 9. Maps of future prediction.**

contributions to saline soil are by wetland and salt pan by 0.1% and 0.5% and negatively by 5% by barren land. Positive contribution in the salt crust is by 0.1% and negatively by vegetation and barren land by 0.2% and 0.24% respectively. Positive contributions to the salt pan by barren land are by 2.79% and negatively by wetland, vegetation, and saline soil. The settlement would experience positively by vegetation, salt pan, barren land, and Aravali hills by 0.18, 0.02, 0.34, and 0.1% respectively and negatively by saline soil by 0.01%. Vegetation will experience positively by salt pan by 0.8% and barren land by 0.7% and negatively by settlement by 0.21% and lastly wetland will experience positively by salt pan by 0.30%, barren land by 0.10% and negatively by saline soil by 0.25%. Spatial cubic trend changes of SSL are given in (Fig 11).

## Discussion

### Loss of saline character

The topmost layer of SSL (0–1.15 m) appears greyish brown to dark grey from top to bottom due to organic residues with minerals of Kieserite providing brine rich in carbonate, sulphate, calcium, magnesium, 1.15–3.6 m appear dark grey with bloedite and brine rich in sodium, 3.6–6 m is rich in calcite, polyhalite, gypsum, dolomite and brine rich in same constituents

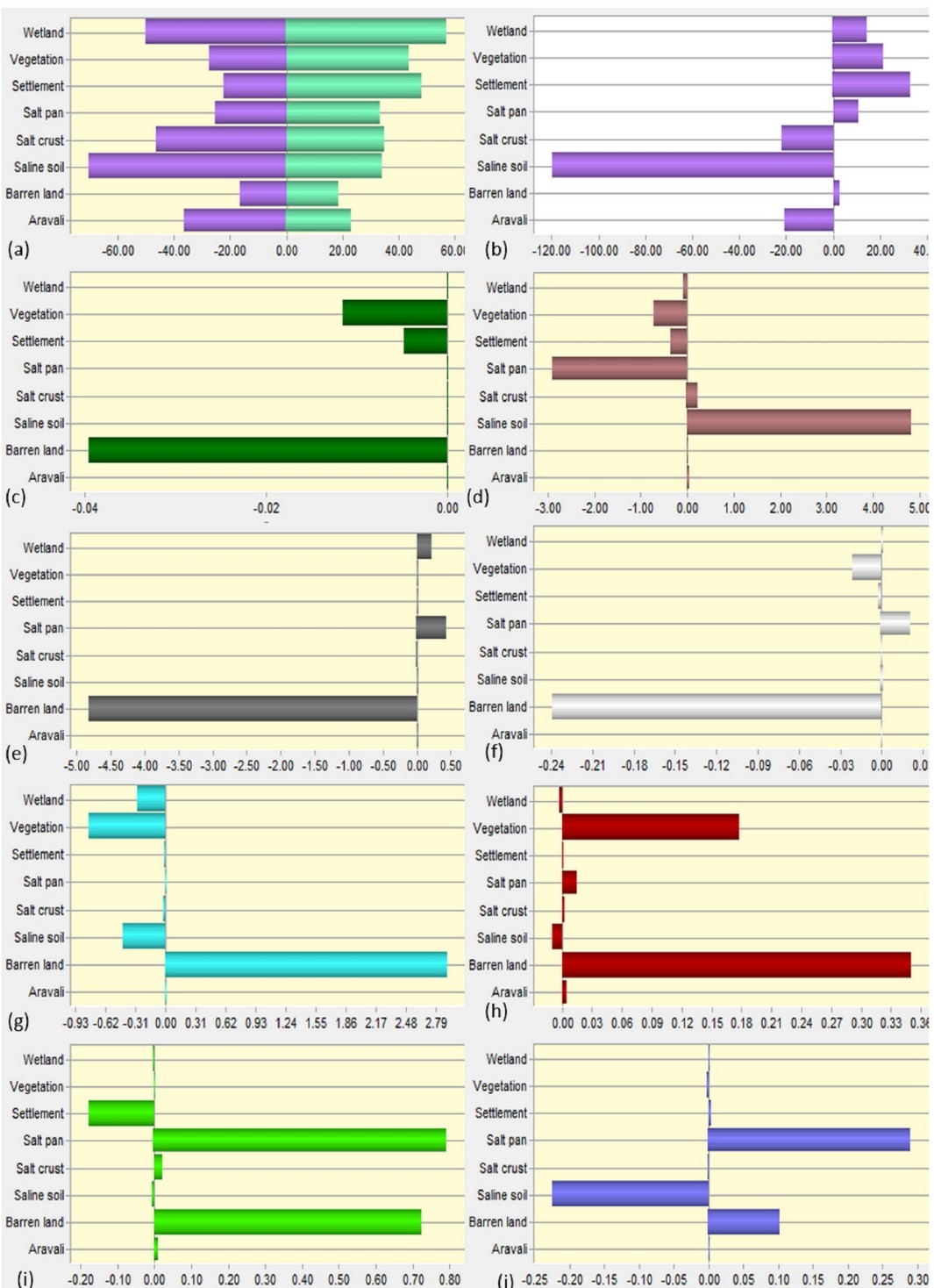

**Fig 10. LULC change percentages.** (a) gain and loss of LULC (b) net change between 2029 and 2059 and (c-j) contributions of each class to net changes.

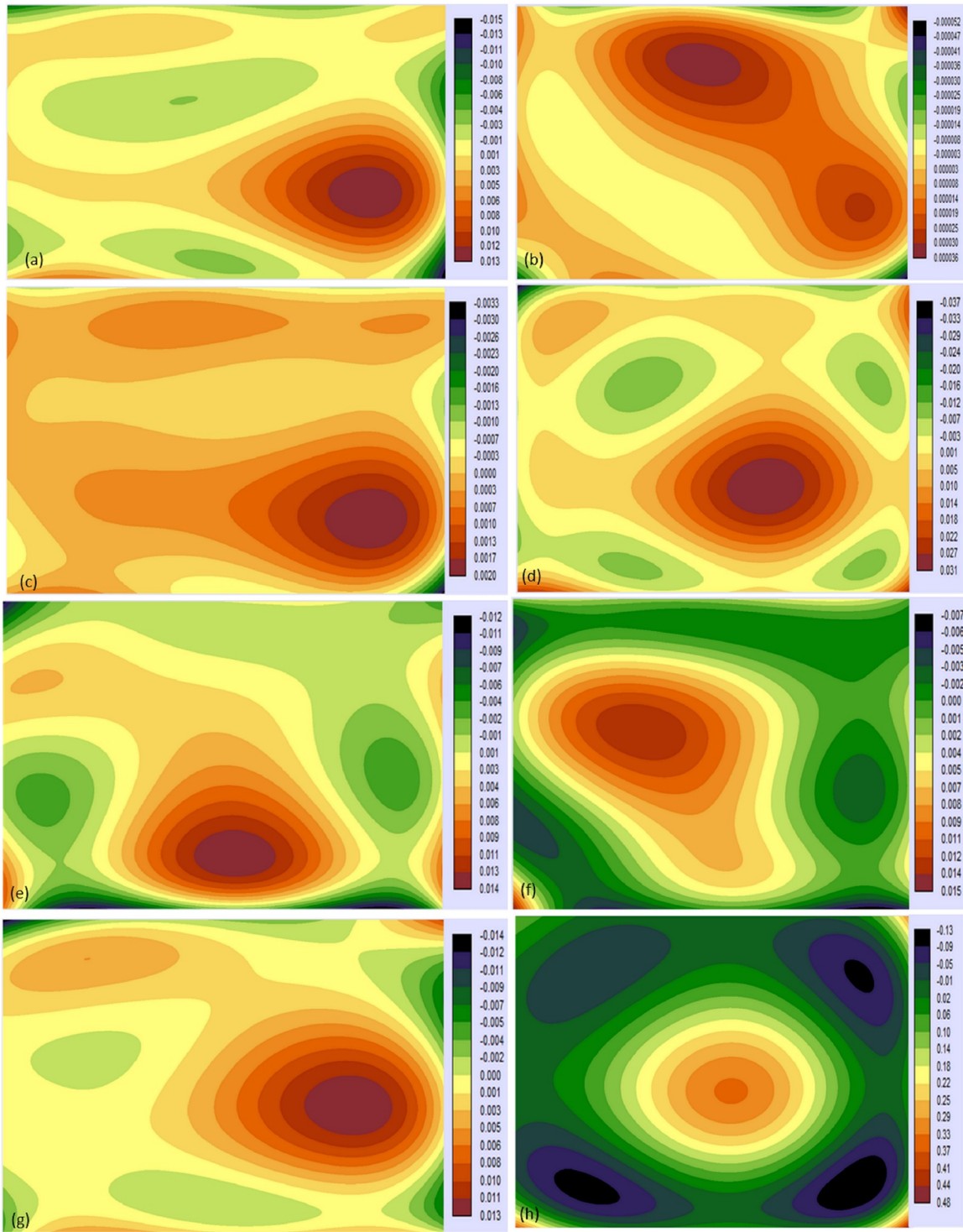

**Fig 11. Spatial trend representation.**

besides potassium, 6–19 m has weathered gypsum, calcite, dolomite, polyhalite, thenardite with no brine and below 19 m has pre-Cambrian rock basement consisting of schists, phyllites and quartzite [19]. However, the decadal analysis states that the six vertical soil gradients are at stake. The only method approved by GoI is the collection of brine through pans and kyars [39]. 18,65,000 tons/year of salt were sustainably extracted in which lake water provided $18^*105$ tons/ year, rainwater $5^*103$ tons/year, and river water $6^*104$ tons/ year [20]. 2 000 illegal tube wells and 240 bore wells have been built over the last two decades. [40] stealing brine worth 300 million USD [39] from both surface and sub-surface [19]. Results from the imagery of 1963 do not suggest that there were unauthorized pans. It occupied 7.4% by authorized Sambhar Salt Ltd. in 1972. Gradually in 1992, 2009 and 2019 encroachment increased to 8.2, 12.3 to 14, 10% respectively. Major encroachment appeared in Nagaur due to the construction of hydrological structures [41]. Other threats are livestock ranching, poaching, sewage discharge, trails [33], vehicle testing [40].

## Loss of wetland connectivity and trophic structure

Within the 3 km SSL buffer zone, Naliasar, Devyani Sarovar, and Ratan Talab are linked by birds for breeding, feeding, and roosting. Their connectivity depends on the water budget, hydrophytes, hydric soil, predator status, food availability, hydro-period, wetland complexes, topography, geography, and weather [42]. However, the results of satellites show a steady decline at 4% from 30.4 to 3.4%. This has forced the bird to move elsewhere. Due to the shrinkage, its complex trophic structure with 39 aquatic and 80 terrestrial producers, 133 primary and secondary consumers [28], and 279 birds as tertiary consumers [21] are at stake. Depending on water availability, level, depth, and microbiota, wetland connectivity is divided into three types; include bottom-dwelling, surface, and shore animals for SSL [26]. Bottom and mud-dwelling animals like *Polypodium sp*. and *Chironomus sp*. survive in favourable seasons from July to December, when salinity is 9.6 to 72.6%, carbon dioxide is in between 48 to 56.2 mg/l, and oxygen is between 42 to 27.8 mg/l [26]. Surface animals consist of both plankton and nekton. Phytoplankton (*Dunaliella saline*, *Aphanotheca halophytica*, *Spirulina sp*) and zooplanktons protozoans, nauplii of crustaceans [26], and nektons are stenohaline that survive during the favourable condition, and replaced by euryhaline animals (*Artemia salina*, *Ephydra macellaria*, and *Eristalis sp*.) during adverse conditions, tolerating up to 164% salinity, and disappear in May to June, when the lake is naturally dry. Shore animals represented by *Labidura riparia*, *Coniocleonus sp*. and others survive during favourable periods; however, they travel to the core during adverse conditions. However, these species might not be available as the lake is shrinking.

## Management and restoration potentials

When saline lakes are relentlessly desiccated, they might become dust bowl that is harmful for both man and environment as in the case of Owens Lake in California or collapse billion-dollar global market of brine shrimp as in case of Lake Urmia or loss of 40,000 metric tons of fishery and 60,000 jobs in Aral Sea [43]. Shrivelled saline lakes create ecological disconnect, neither support unique halophytes nor attract flamingos or other birds [44]. So, there is a pressing need that the Sambhar Lake is given urgent attention. If not then it requires more capital for restoration than the revenue it generated as in the case of Owen's Lake for which US$ 3.6 billion was spent for its dust mitigation [45]. At the current stage, it is possible through the reconstruction of its physicochemical adjustments, and reintroduction of native flora and fauna. Emphasis on health protection, incentives, and rewards must be given to the salt workers so that more people will participate in the wise use of this lake. Demolishing check dams

and anicuts, ban of sub-surface brine collection, using electrical pumps, illegal salt pan encroachment in and around lake be declared as a punishable act, demolish construction up to 3 km buffer zone declaring it 'no construction zone', controlled sewage disposal, increasing water residence period. Increasing aquatic biodiversity, hydrodynamics, nutrient cycling, vegetative and non- vegetative productivity, cascading trophic levels be focused. These steps will not only help SSL to restore its pristine conditions but also generate revenue for a longer period, provide jobs to more people, and attract more migratory birds.

## Conclusion

In the present article, we identified the key drivers affecting the largest inland saline Ramsar site in the semi-arid region of India over 10 decades (1963–2059) at a landscape level. We used the CA-Markov model using the geospatial platform with the ground observations for the identification. The results show the reduction of wetlands from 30.7 to 3.4% from 1972 till 2019. subsequently increasing barren land by 4.2%; salt pans by 6.6% and settlement by 1.2% till 2019. Likewise, future prediction shows loss of 40% wetland and 120% of saline soil and net increase in 30% vegetation, 40% settlement, 10% salt pan, 5% barren land, and a net loss of 20%, each by Aravali hills and salt crust. The major drivers are illegal salt pan encroachment, excess groundwater extraction, increasing settlement area, and water diversion. Taken together, our findings and the findings of previous studies point towards its complete desiccation towards a wasteland. Subsequently, this will not be able to generate billion dollars of revenue, or attract lakhs of migratory birds or provide jobs to thousands of salt workers. Our findings offer a novel restoration strategy which could be of primary interest for the next decade of UN Decade on Ecosystem Restoration.

## Supporting information

**S1 Protocol. Step by step protocol.**
(PDF)

**S2 Protocol.**
(DOCX)

**S1 Dataset. All the raw and processed dataset.**
(XLSX)

## Acknowledgments

We thank volunteers of Asian Waterbird Census, Wetlands International, South Asia and the members of Wildlife Creatures Organization (WCO), Phulera for the constant help during the field visits and bird census required for this study. We are also grateful to the editors and anonymous reviewers for their vital comments that improved the manuscript.

## Author Contributions

**Conceptualization:** Rajashree Naik, Laxmikant Sharma.

**Data curation:** Rajashree Naik, Laxmikant Sharma.

**Formal analysis:** Laxmikant Sharma.

**Investigation:** Rajashree Naik, Laxmikant Sharma.

**Methodology:** Rajashree Naik, Laxmikant Sharma.

**Resources:** Laxmikant Sharma.

**Software:** Rajashree Naik, Laxmikant Sharma.

**Supervision:** Laxmikant Sharma.

**Validation:** Rajashree Naik.

**Visualization:** Laxmikant Sharma.

**Writing – original draft:** Rajashree Naik.

**Writing – review & editing:** Laxmikant Sharma.

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
