## [Decision Letter · Decision Letter 0]

10 May 2021

PONE-D-21-05894

Spatio-temporal modelling for the evaluation of an altered Indian saline Ramsar site and its drivers for ecosystem management and restoration

PLOS ONE

Dear Dr. LAXMIKANT SHARMA,

Thank you for submitting your manuscript to PLOS ONE. After careful consideration, we feel that it has merit but does not fully meet PLOS ONE’s publication criteria as it currently stands. Therefore, we invite you to submit a revised version of the manuscript that addresses the points raised during the review process.

We look forward to receiving your revised manuscript.

Kind regards,

Bijeesh Kozhikkodan Veettil

Academic Editor

PLOS ONE

Journal Requirements:

3. In your Methods section, please provide additional location information of the study sites, including geographic coordinates for the data set if available.

6. We note that Figure 1, 3, 5, 7, 8, 9 in your submission contain map images which may be copyrighted. All PLOS content is published under the Creative Commons Attribution License (CC BY 4.0), which means that the manuscript, images, and Supporting Information files will be freely available online, and any third party is permitted to access, download, copy, distribute, and use these materials in any way, even commercially, with proper attribution. For these reasons, we cannot publish previously copyrighted maps or satellite images created using proprietary data, such as Google software (Google Maps, Street View, and Earth). For more information, see our copyright guidelines: http://journals.plos.org/plosone/s/licenses-and-copyright.

6.1.    You may seek permission from the original copyright holder of Figure 1, 3, 5, 7, 8, 9 to publish the content specifically under the CC BY 4.0 license. 

Reviewers' comments:

Reviewer's Responses to Questions

**Comments to the Author**

1. Is the manuscript technically sound, and do the data support the conclusions?

Reviewer #1: Yes

Reviewer #2: Yes

2. Has the statistical analysis been performed appropriately and rigorously? 

Reviewer #1: Yes

Reviewer #2: No

3. Have the authors made all data underlying the findings in their manuscript fully available?

Reviewer #1: Yes

Reviewer #2: Yes

4. Is the manuscript presented in an intelligible fashion and written in standard English?

Reviewer #1: Yes

Reviewer #2: Yes

5. Review Comments to the Author

Reviewer #1: Please see attached marked up manuscript and figure files.

I feel that the paper has the merit and nicely written. The comments has been raised in the attached ms itself. Figures require lot of changes and additions for clarity.

Once taken care, the paper may be considered for publication.

Reviewer #2: There are some significant issues in this manuscript enlisted as below:

a. Abstract need to be highlighted with the achieved experimental outcome.

b. In the Introduction section, authors need to high "What is the main contribution of the manuscript", "What problem is identified from the related work used in the manuscript" and "explained the structures of manuscript"

c. There is introductory lines between the section and sub-section

d. Figure 2, Methodology, What technique is used to process the data? and What problems identified by the authors that can be solved by data preprocessing techniques?

e. All figures used in this manuscript are very low quality, it need to replace with the 300 dpi images.

f. To analysis the results, authors used the kappa coefficient. There are more statistical paramaters, authors should also consider that paramaters.

g. What is the evaluation testbed for the experimental analysis? Which kind of software is used to achieve that output. It is more benificial for the other researchers.

h. Conclusion section need to be partially modified with experimental outcome from the study ?

g. There are some typo errors in the whole manuscript. Authors need to check and revise that mistakes.

6. PLOS authors have the option to publish the peer review history of their article (what does this mean?). If published, this will include your full peer review and any attached files.

Reviewer #1: No

Reviewer #2: **Yes: **Vikram Puri

---

## [Author Response · Author response to Decision Letter 0]

15 Jun 2021

Dear Reviewers,

We deeply express our gratitude towards your continuous and tiring efforts to read, understand, analyze our manuscript with id “PONE-D-21-05894” and provide suggestion. Your noble suggestions have tremendously improved the quality of the work. The suggestion for graphics included modification of study area map, Land Use Land Cover maps, soil and water quality maps, field visit maps, research methodology flowchart and graphs which were highly creditable. Further, suggestion for writeup included modification of abstract, introduction, correction of figure numbering, experimental testbeds, and processing techniques for satellite images. 

All the suggestions have been accepted and incorporated to the best of our knowledge. We have segregated the comments according to the reviewers and provided the details of each action taken, and their location in the manuscript in a tabular form provided in the next section.

Once again, we humbly thank you for the strenuous efforts you have taken and for the patience to receive our response. We look forward to hear from you regarding our resubmission. We would be glad to respond to any further questions and comments that you may have.

---

## [Decision Letter · Decision Letter 1]

12 Jul 2021

Spatio-temporal modelling for the evaluation of an altered Indian saline Ramsar site and its drivers for ecosystem management and restoration

PONE-D-21-05894R1

Dear Dr. Sharma,

We’re pleased to inform you that your manuscript has been judged scientifically suitable for publication and will be formally accepted for publication once it meets all outstanding technical requirements.

Kind regards,

Bijeesh Kozhikkodan Veettil

Academic Editor

PLOS ONE

Additional Editor Comments (optional):

Reviewers' comments:

Reviewer's Responses to Questions

**Comments to the Author**

1. If the authors have adequately addressed your comments raised in a previous round of review and you feel that this manuscript is now acceptable for publication, you may indicate that here to bypass the “Comments to the Author” section, enter your conflict of interest statement in the “Confidential to Editor” section, and submit your "Accept" recommendation.

Reviewer #1: All comments have been addressed

Reviewer #2: All comments have been addressed

2. Is the manuscript technically sound, and do the data support the conclusions?

Reviewer #1: Yes

Reviewer #2: Yes

3. Has the statistical analysis been performed appropriately and rigorously? 

Reviewer #1: Yes

Reviewer #2: Yes

4. Have the authors made all data underlying the findings in their manuscript fully available?

Reviewer #1: Yes

Reviewer #2: Yes

5. Is the manuscript presented in an intelligible fashion and written in standard English?

Reviewer #1: Yes

Reviewer #2: Yes

6. Review Comments to the Author

Reviewer #1: All the suggestions and comments raised in the earlier version has been made by the authors and also they clarified the queries.

Reviewer #2: (No Response)

7. PLOS authors have the option to publish the peer review history of their article (what does this mean?). If published, this will include your full peer review and any attached files.

Reviewer #1: No

Reviewer #2: No

---

## [Editor Report · Acceptance letter]

14 Jul 2021

PONE-D-21-05894R1 

Spatio-temporal modelling for the evaluation of an altered Indian saline Ramsar site and its drivers for ecosystem management and restoration 

Dear Dr. Sharma:

I'm pleased to inform you that your manuscript has been deemed suitable for publication in PLOS ONE. Congratulations! Your manuscript is now with our production department. 

Kind regards, 

on behalf of

Dr. Bijeesh Kozhikkodan Veettil 

Academic Editor

PLOS ONE